# Attacking Graph Convolutional Networks via Rewiring

## Abstract

Graph Neural Networks (GNNs) have boosted the performance of many graph related tasks such as node classification and graph classification. Recent researches show that graph neural networks are vulnerable to adversarial attacks, which deliberately add carefully created unnoticeable perturbation to the graph structure. The perturbation is usually created by adding/deleting a few edges, which might be noticeable even when the number of edges modified is small. In this paper, we propose a graph rewiring operation which affects the graph in a less noticeable way compared to existing operators. We then use reinforcement learning to learn the attack strategy based on the proposed rewiring operation. Experiments on real world graphs demonstrate the effectiveness of the proposed framework. To understand the proposed framework, we further analyze how its generated perturbation to the graph structure affects the output of the target model and the advantages of the rewiring operation.

## 1 Introduction

Graph structured data are ubiquitous in many real world applications. Various data from different domains, such as social networks, molecular graphs and transportation networks can all be modeled as graphs. Recently, increasing effort has been devoted towards developing deep neural networks on graph structured data. This stream of works, which is known as Graph Neural Networks (GNN) has shown to enhance the performance in many graph related tasks such as node classification (Kipf & Welling, 2016; Hamilton et al., 2017) and graph classification (Bruna et al., 2013; Defferrard et al., 2016; Ying et al., 2018; Zhang et al., 2018).

Recent researches have shown that deep neural networks are highly vulnerable to adversarial attacks (Szegedy et al., 2013; Goodfellow et al., 2014; Kurakin et al., 2016; Carlini & Wagner, 2017). In computer vision, performing an adversarial attack is to add deliberately created, but unnoticeable, perturbation to a given image such that the deep model misclassifies the perturbed image. Unlike image data, which can be represented in the continuous space, graph structured data is discrete. Few efforts have been made to investigate the robustness of graph neural networks against adversarial attacks. Only recently, such researches about adversarial attacks on graph structured data started to emerge. A greedy algorithm is proposed to attack the semi-supervised node classification task (Zügner et al., 2018). The method deliberately tries to modify the graph structure and node features such that the label of a targeted node can be changed. A reinforcement learning based algorithm is proposed to attack both node classification and graph classification task by only modifying the graph structure(Dai et al., 2018). A meta-learning based attack method is designed to impair the overall performance of the node classification task(Zügner & Günnemann, 2019). For the majority of existing works, the graph structure is modified by adding or deleting edges.

To ensure that the difference between the attacked graph and the original graph is "unnoticeable", the number of actions (adding/deleting edges) that can be taken by the attacking algorithms is usually constrained by a budget. However, even when this budget is small, adding or deleting edges can still make "noticeable" changes to the graph structure (Miller et al., 2019). For example, it is evident that many important graph properties are based on eigenvalues and eigenvectors of the Laplacian matrix of the graph (Chan & Akoglu, 2016); while adding or deleting an edge can make remarkable changes on the eigenvalues/eigenvectors of the graph Laplacian (Ghosh & Boyd, 2006). Thus, in this work, we propose a new operation based on graph rewiring. A single rewiring operation involves

three nodes $(v_{fir}, v_{sec}, v_{thi})$, where we remove the existing edge between $v_{fir}$ and $v_{sec}$ and add edge between $v_{fir}$ and $v_{thi}$. Note that $v_{thi}$ is constraint to be the 2-hop neighbor of $v_{fir}$ in our setting. It is obvious that the proposed rewiring operation preserves some basic properties of the graph such as number of nodes and edges, total degrees of the graph and etc, while operations like adding and deleting edges cannot. Furthermore, the proposed rewiring operation affects some of the important measures based on graph Laplacian such as algebraic connectivity in a smaller way than adding/deleting edges, which is theoretically demonstrated in Section 4.1. In addition, the rewiring operation is a more natural way to modify the graph. For example, in biology, the evolution of DNA and amino acid sequences can lead to pervasive rewiring of protein–protein interactions (Zitnik et al., 2019).

In this paper, we aim to construct adversarial examples by performing rewiring operations for the task of graph classification. More specifically, we treat the process of applying a series of rewiring operations to a given graph as a discrete Markov decision process (MDP) and use reinforcement learning to learn how to make these decisions. We demonstrate the effectiveness of the proposed algorithm on real-world graphs. Then we further analyze how the adversarial changes in the graph structure affect both the graph embedding learned by the graph neural network model and the output label and illustrate the advantages of the rewiring operation.

## 2 BACKGROUND

In this section, we introduce notations and the target graph convolutional model we seek to attack. We denote a graph as $G = \{\mathcal{V}, \mathcal{E}\}$, where $\mathcal{V} = \{v_1, \ldots, v_{|\mathcal{V}|}\}$ and $\mathcal{E} = \{e_1, \ldots, e_{|\mathcal{E}|}\}$ are the sets of nodes and edges, respectively. The edges describe the relations between nodes, which can be described by an adjacency matrix $\mathbf{A} \in \{0, 1\}^{|\mathcal{V}| \times |\mathcal{V}|}$. $\mathbf{A}_{ij} = 1$ means $v_i$ and $v_j$ are connected, 0 otherwise. Each node in the graph has some features that are associated with it. These features are represented as a matrix $\mathbf{X} \in \mathbb{R}^{|\mathcal{V}| \times d}$, where the $i$-th row of $\mathbf{X}$ denotes the node features of node $v_i$ and $d$ is the dimension of features. Thus, an attributed graph can be represented as $G = \{\mathbf{A}, \mathbf{X}\}$.

### 2.1 GRAPH CLASSIFICATION

In the setting of graph classification, we are given a set of graphs $\mathcal{G} = \{G_i\}$. Each of these graphs $G_i$ is associated with a label $y_i$. The task is to build a good classifier using the given set of graphs such that it can make correct predictions when new unseen graphs are fed into it. A graph classifier parameterized by $\theta$ can be represented as $f(G|\theta) = y^o$, where $y^o$ denotes the label of a graph $G \in \mathcal{G}$ predicted by the classifier. The parameters $\theta$ in the classifier $f(\cdot|\theta)$ can be learned by solving the following optimization problem $\min_\theta \sum_i L(f(G_i|\theta), y_i)$, where $L(\cdot, \cdot)$ is used to measure the difference between the predicted and ground truth labels. Cross entropy is a commonly adopted measurement for $L(\cdot, \cdot)$.

### 2.2 GRAPH CONVOLUTION NETWORKS

Recently, Graph Neural Networks have been shown to be effective in graph representation learning. These models usually learn node representations by iteratively aggregating, transforming and propagating node information. In this work, we adopt the graph convolutional networks (GCN) (Kipf & Welling, 2016). A graph convolutional layer in the GCN framework can be represented as

$$\mathbf{F}^j = ReLU(\mathbf{D}^{-\frac{1}{2}} \mathbf{A} \mathbf{D}^{-\frac{1}{2}} \mathbf{F}^{j-1} \mathbf{W}^j) \tag{1}$$

where $\mathbf{F}^j \in \mathbb{R}^{N \times d_j}$ is the output of the $j$-th layer and $\mathbf{W}^j$ represents the parameters of this layer. A GCN model usually consists of $J$ graph convolutional layers, with $\mathbf{F}^0 = \mathbf{X}$. The output of the GCN model is $\mathbf{F}^J$, which is denote as $\mathbf{F}$ for convenience. To obtain a graph level embedding $\mathbf{u}_G$ for graph $G$ to perform graph classification, we apply a global pooling over the node embeddings.

$$\mathbf{u}_G = pool(\mathbf{F}) \tag{2}$$

Different global pooling functions can be used, and we adopt the max pooling in this work. A multilayer perceptron (MLP) and softmax layer are then sequentially applied on the graph embedding to predict the label of the graph

$$y^o = \operatorname{argmax} softmax(MLP(\mathbf{u}_G | \mathbf{W}_{MLP})) \tag{3}$$

where $MLP(\cdot|\mathbf{W}_{MLP})$ denotes the multilayer perceptron with parameters as $\mathbf{W}_{MLP}$. A GCN-based classifier for graph classification can be described using eq. equation 1, equation 2 and equation 3 as introduced above. For simplicity, we summarize it as $y^o = f_{GCN}(G|\theta_{GCN})$, where $\theta_{GCN}$ includes all the parameters in the model.

## 3 PROBLEM FORMULATION

In this work, we aim to build an attacker $\mathcal{T}$ that takes a graph as input and modify the structure of the graph to fool a GCN classifier. Modifying a graph structure is equivalent to modify its adjacency matrix. The function of the attacker can be represented as $\tilde{G} = \mathcal{T}(G) = \{\mathcal{T}(\mathbf{A}), \mathbf{X}\} = \{\tilde{\mathbf{A}}, \mathbf{X}\}$. Given a classifier $f(\cdot)$, the goal of the attacker is to modify the graph structure so that the classifier outputs a different label from what it originally predicted. Note here, we neglect the $\theta$ inside $f(\cdot)$, as the classifier is already trained and fixed. Mathematically, the goal of the attacker can be represented as: $f(\mathcal{T}(G)) \neq f(G)$.

As described above, the attacker $\mathcal{T}$ is specifically designed for a given classifier $f(\cdot)$. To reflect this in the notation, we now denote the attacker for the classifier $f(\cdot)$ as $\mathcal{T}_f$. In our work, the attacker $\mathcal{T}_f$ has limited knowledge of the classifier. The only information the attacker can get from the classifier is the label of (modified) graphs. In other words, the classifier $f(\cdot)$ is treated as a black-box model for the attacker $\mathcal{T}_f$.

An important constraint to the attacker $\mathcal{T}_f$ is that it is only allowed to make "unnoticeble" changes to the graph structure. To account for this, we propose the *rewiring* operation, which is supposed to make more subtle changes than adding or deleting edges. We will show that the rewiring operation can better preserve a lot of important properties of the graph compared to adding or deleting edges in Section 4.1. We also empirically compare the rewiring operation with the deleting/adding edges in Appendix C. The definition of the proposed rewiring is given below:

**Definition 1.** *A rewiring operation* $\mathbf{a}$ *involves three nodes and it can be denoted as* $\mathbf{a} = \{v_{fir}, v_{sec}, v_{thi}\}$*, where* $v_{sec} \in N^1(v_{fir})$ *and* $v_{thi} \in N^2(v_{fir})/N^1(v_{fir})$*.* $N^k(v_{fir})$ *denotes the k-th hop neighbors of* $v_{fir}$ *and the sign* / *stands for exclusion. The rewiring operation deletes the existing edge between nodes* $v_{fir}$ *and* $v_{sec}$*, while adding an edge to connect nodes* $v_{fir}$ *and* $v_{thi}$*.*

The attacker $\mathcal{T}_f$ is given a budget of $K$ proposed rewiring operations to modify the graph structure. A straightforward way to set $K$ is choosing a small fix number. However, it is likely that graphs in a given data set have various graph sizes. The same number of rewiring operations can affect the graphs of different size in various magnitude. Hence, a more suitable way is to allow flexible number of rewiring operations according to the graph size. Thus, we propose to use $K = p \cdot |\mathcal{E}|$ for a given graph $G$, where $p \in (0, 1)$ is a ratio.

The process of the attacker on a graph $G$ can be now denoted as $\mathcal{T}_f(G) \leftrightarrow (a_1, a_2, \ldots, a_M)[G]$, where the right hand part means to sequentially apply the rewiring operations $a_1, \ldots, a_M$ to the graph $G$, and $M$ is the number of rewiring operations taken with $M \leq K$.

## 4 REWIRING-BASED ATTACK TO GRAPH CONVOLUTIONAL NETWORKS

Next, we first discuss the properties of the proposed rewiring operation to show its advantages. We then introduce the proposed attacking framework ReWatt based on reinforcement learning and rewiring.

### 4.1 PROPERTIES OF THE PROPOSED REWIRING OPERATION

The proposed rewiring operation has several advantages compared to simply adding or deleting edges. More empirical discussions can be found in Appendix C. One obvious advantage of the proposed rewiring operation is that it does not change the number of nodes, the number of edges and the total degree of a graph. However, operations like "adding" or "deleting" edges may change those properties.

Many important graph properties are based on the eigenvalues of the Laplacian matrix of a graph (Chan & Akoglu, 2016) such as Algebraic Connectivity Fiedler (1973) and Effective Graph

Resistance Ellens et al. (2011). A detailed description of Algebraic Connectivity and Effective Graph Resistance are given in Appendix A. Next, we demonstrate that the proposed rewiring operation is likely to make smaller changes to eigenvalues, which result in unnoticeable changes under graph Laplacian based measures. For a graph $G$ with $\mathbf{A}$ as its adjacency matrix, its Laplacian matrix $\mathbf{L}$ is defined as $\mathbf{L} = \mathbf{D} - \mathbf{A}$, where $\mathbf{D}$ is the diagonal degree matrix (Mohar et al., 1991). Let $\lambda_1, \ldots, \lambda_{|\mathcal{V}|}$ denote the eigenvalues of the Laplacian matrix arranged in the increasing order with $\mathbf{x}_1, \ldots, \mathbf{x}_{|\mathcal{V}|}$ being the corresponding eigenvectors. We show how a single proposed rewiring operation affects the eigenvalues. Our analysis is based on the following lemma:

**Lemma 1.** *(Stewart, 1990) Let $(\alpha_i, \mathbf{h}_i)$ be the eigen-pairs of a symmetric matrix $\mathbf{M} \in \mathbb{R}^{N \times N}$. Given a perturbation $\Delta \mathbf{M}$ to matrix $\mathbf{M}$, its eigenvalues can be updated by $\Delta \alpha_i = \mathbf{h}_i^T \Delta \mathbf{M} \mathbf{h}_i$.*

The proof can be found in (Stewart, 1990). Using this lemma, we have the following corollary

**Corollary 1.** *For a given graph $G$ with Laplacian matrix $\mathbf{L}$, one proposed rewiring operation $(v_{fir}, v_{sec}, v_{thi})$ affects the eigen-value $\lambda_i$ by $\Delta \lambda_i$, for $i = 1, \ldots, |\mathcal{V}|$, where*

$$\Delta \lambda_i = (2\mathbf{x}_i[fir] - \mathbf{x}_i[thi] - \mathbf{x}_i[sec])(\mathbf{x}_i[sec] - \mathbf{x}_i[thi]) \tag{4}$$

*where $\mathbf{x}_i[index]$ denotes the $index$-th value of the eigenvector $\mathbf{x}_i$.*

The proof can be found in Appendix B.

Furthermore, each eigenvalue $\lambda_i$ of the Laplacian matrix measures the "smoothness" of its corresponding eigenvector $\mathbf{x}_i$ (Shuman et al., 2012; Sandryhaila & Moura, 2014). The "smoothness" of an eigenvector measures how different its elements are from their neighboring nodes. Thus, the first few eigenvectors with relatively small eigenvalues are rather "smooth". Note that in the proposed rewiring operation, $v_{sec}$ is the direct neighbor of $v_{fir}$ and $v_{thi}$ is the 2-hop neighbor of $v_{fir}$. Thus, the difference $\mathbf{x}_i[fir] - \mathbf{x}_i[thi]$ is expected to be smaller than the difference $\mathbf{x}_i[fir] - \mathbf{x}_i[can]$, where $\mathbf{x}_i[can]$ can be any other node that is further away. This means that the proposed rewiring operation (to 2-hop neighbors) is likely to make smaller changes to the first few eigenvalues than rewiring to any further away nodes or adding an edge between two nodes that are far away from each other.

## 4.2 GRAPH ADVERSARIAL ATTACK WITH REINFORCEMENT LEARNING

Given a graph $G$, the process of the attacker $\mathcal{T}$ is a general decision making process $M = (\mathcal{S}, \mathcal{A}, P, R)$, where $\mathcal{A} = \{a_t\}$ is the set of actions, which consists of all valid rewiring operations, $\mathcal{S} = \{s_t\}$ is the set of states that consists of all possible intermediate and final graphs after rewiring, $P$ is the transition dynamics that describes how a rewiring action $a_t$ changes the graph structure $p(s_{t+1}|, s_t, \ldots, s_1, a_t)$. $R$ is the reward function, which gives the reward for the action taken at a given state. Thus, the procedure of attacking a graph can be described by a trajectory $(s_1, a_1, r_1, \ldots, s_M, a_M, r_M)$, where $s_1 = G$. The key point for the attacker is to learn how to make the decision of picking a suitable rewiring action when at the state $s_t$. This can be done by learning a policy network to get the probability $p(a_t|s_t, \ldots, s_1)$ and sample the rewiring operation correspondingly. Modelling in this way, the decision making at a state $s_t$ is dependant on all its previous states, which could be difficult to model due to the long-term dependency. It is easy to notice that the intermediate states $s_t$ are all predicted to have the same label as the original graph. Thus, we can treat each of the states as a brand new graph to be attacked regardless of what leads to it. That is to say, the decision making at the state $s_t$ can be solely dependant on the current state, $p(a_t|s_t, \ldots, s_1) = p(a_t|s_t)$. Thus, we model the process of attack as a Markov Decision Process (MDP) Sutton & Barto (2018). Hence, we adopt reinforcement learning to learn how to make effective decisions. We name the proposed framework as ReWatt. The key elements of the environment for the reinforcement learning are defined as follows:

**State Space** The state space of the environment consists of all the intermediate graphs generated after the possible rewiring operations.

**Action Space** The action space consists of the valid rewiring operations as defined in Definition 1.

**State Transition Dynamics** Given an action (rewiring operation) $a_t = \{v_{fir}, v_{sec}, v_{thi}\}$ at state $s_t$. The next state $s_{t+1}$ is achieved by deleting the edge between $v_{fir}$ and $v_{sec}$ in the current state $s_t$ and adding an edge to connect $v_{fir}$ with $v_{thi}$.

**Reward Design** The main goal of the attacker is to make the classifier $f(\cdot)$ predict a different label than originally predicted. We also encourage the attacker to take as few actions as possible so that

the modification to the graph structure is minimal. Thus, we assign a positive reward when the attack is successful and assign a negative reward for each action step taken. The reward $R(s_t, a_t)$ is given as

$$R(s_t, a_t) = \begin{cases} 1 & \text{if } f(s_t) \neq f(s_1); \\ n_r & \text{if } f(s_t) = f(s_1). \end{cases}$$

where $n_r$ is the negative reward to penalize each step taken. Similar to how we set a flexible rewiring budget $K$, here we propose to use $n_r = -\frac{1}{K} = -\frac{1}{p \cdot |\mathcal{E}|}$, which depends on the size of the graph.

**Termination** The attack process will stop either when the number of actions reaches the budget $K$ or the attacker successfully "changed" the label of the slightly modified graph.

## 4.3 POLICY NETWORK

In this subsection, we introduce the policy network to learn the policy $p(a_t|s_t)$ on top of the graph representations learned by GCN. However, this GCN is different from the target classifier one, since it has 2 convectional layers. To choose a valid proposed rewiring action, we decompose the rewiring action to 3 steps: 1) choosing an edge $e_t = (v_{e_1}, v_{e_2})$ from the set of edges of the intermediate graph $s_t$; 2) determining $v_{e_{t_1}}$ or $v_{e_{t_2}}$ to be $v_{fir_t}$ and the other to be $v_{sec_t}$; and 3) choosing the third node $v_{thi_t}$ from $N_{s_t}^2(v_{fir_t})/N_{s_t}^1(v_{fir_t})$. Correspondingly, we decompose $p(a_t|s_t)$ as follows

$$p(a_t|s_t) = p_{edge}(e_t|s_t) \cdot p_{fir}(v_{fir_t}|e_t, s_t) \cdot p_{thi}(v_{thi_t}|v_{fir_t}, e_t, s_t) \tag{5}$$

We design three policy networks based on GCN to estimate the three distributions in the right hand of the equation equation 5, which will be introduced next. To select an edge from the edge set $\mathcal{E}_{s_t}$, we generate the edge representation from the node representations $\mathbf{F}_{s_t} \in \mathbb{R}^{|\mathcal{V}_{s_t}| \times d_F}$ learned by GCN. For an edge $e = (v_{e_1}, v_{e_2})$, the edge representation can be represented as $\mathbf{e} = concat(\mathbf{u}_{s_t}, h(\mathbf{F}_{s_t}[e_1, :], \mathbf{F}_{s_t}[e_2, :]))$, where $\mathbf{u}_{s_t}$ is the graph representation of the state $s_t$, $h(\cdot, \cdot)$ is a function to combine the two node representations and $concat(\cdot, \cdot)$ denotes the concatenation operation. We include $\mathbf{u}_{st}$ in the representation of the edge to incorporate the graph information when making the decision. The representation of all the edges in $\mathcal{E}_{s_t}$ can be represented as a matrix $\mathbf{E}_{s_t} \in \mathbb{R}^{|\mathcal{E}_{s_t}| \times 2d_F}$, where each row represents an edge. The probability distribution over all the edges can be represented as

$$p_{edge}(\cdot|s_t) = softmax(MLP(\mathbf{E}_{s_t}|\theta_{edge})), \tag{6}$$

where we use $MLP(\cdot|\theta_{edge})$ to denote a Multilayer Perceptron that maps $\mathbf{E}_{s_t} \in \mathbb{R}^{|\mathcal{E}_{s_t}| \times 2d_F}$ to a vector in $\mathbb{R}^{|\mathcal{E}_{s_t}|}$, which, after going through the softmax layer, represents the probability of choosing each edge. Let $e_t = (v_{e_{t_1}}, v_{e_{t_2}})$ denote the edge sampled according to eq. equation 6. To decide which node is going to be the first node, we estimate the probability distribution over these two nodes as

$$p_{fir}(\cdot|e_t, s_t) = softmax(MLP([\mathbf{v}_{e_{t_1}}, \mathbf{v}_{e_{t_2}}]^T|\theta_{fir})) \tag{7}$$

where $\mathbf{v}_{e_{t_i}} = concat(\mathbf{e}_t, \mathbf{F}_{s_t}[e_{t_i}, :]) \in \mathbb{R}^{3d_F}$ for $i = 1, 2$. The first node can be sampled from the two nodes $v_{e_{t_1}}, v_{e_{t_2}}$ according to eq. equation 7. We then proceed to estimate the probability distribution $p(\cdot|v_{fir_t}, e_t, s_t)$. For any node $v_c \in N^2(v_{fir_t})/N^1(v_{fir_t})$, we use $\hat{\mathbf{v}}_c = concat(\mathbf{v}_{e_{t_1}}, \mathbf{F}_{s_t}[c, :])$ to represent it. The representations for all the nodes in $N^2(v_{fir_t})/N^1(v_{fir_t})$ can be represented by a matrix $\hat{\mathbf{V}}_{s_t} \in \mathbb{R}^{|N^2(v_{fir_t})/N^1(v_{fir_t})| \times 4d_F}$ with each row representing a node. The probability distribution of choosing the third node over all the candidate nodes can be modeled as:

$$p_{thi}(\cdot|v_{fir_t}, e_t, s_t) = softmax(MLP(\hat{\mathbf{V}}_{s_t}|\theta_{thi})) \tag{8}$$

The third node $v_{thi_t}$ can be sampled from the set of candidate nodes $N^2(v_{fir_t})/N^1(v_{fir_t})$ according to the probability distribution in eq equation 8. An action $a_t$ can be generated by sequentially estimating and sampling from the probability distributions in eq. equation 6, equation 7 and equation 8.

## 4.4 PROPOSED FRAMEWORK - REWATT

With the rewiring and the policy network defined above, our overall framework can be summarized as follows. With State $s_t$, the Attacker uses GCN to learn node and edge embeddings, which are

used as input to Policy Networks to make decision about the next action. Once the new action is sampled from the policy network, rewiring is performed on $s_t$ and we arrive in the new state $s_{t+1}$. We query the black-box classifier to get the prediction $f(S_{t+1})$, which is compared with $f(s_1)$ to get reward. Policy gradient (Sutton & Barto, 2018) is adopted to learn the policies by maximizing the rewards.

## 5 EXPERIMENT

In this section, we conduct experiments to evaluate the performance of the proposed framework ReWatt. We also carry out a study to analyze how the trained attacker works. Some empirical investigation on the advancements of the rewiring operation can be found in Appendix C.

### 5.1 ATTACK PERFORMANCE

To demonstrate the effectiveness of ReWatt, we conduct experiments on three widely used social network data sets (Kersting et al., 2016) for graph classification, i.e., REDDIT-MULTI-12K, REDDIT-MULTI-5K and IMDB-MULTI (Yanardag & Vishwanathan, 2015). The statistics can be found in Appendix D. Note that the re-wiring operation (as well as the other operations) may lead to abnormal structure of some kinds of graphs, which can make the graphs invalid, especially for chemical molecules. So in this paper, We avoid chemical related datasets but only use social networks datasets. In the social domain, if the changes are subtle, it is most likely that we will not we will not introduce abnormal structures. Meanwhile, it is straightforward to extend our framework to datsets from the other domains if we have expertise in them. For example, if we know what structures are abnormal, we can use such knowledge to constraint the the state space of the RL framework. We leave it as one future work.

In this work, the classifier we target to attack is the GCN-based classifier as introduced in Section 2. We set the number of layers to 3. We need to train the classifier using a fraction of the data and then treat the classifier as a black box to be attacked. We then use part of the remaining data to train the attacker and use the rest of the data to test the performance of the attacker. Thus, for each data set, we split it into three parts with the ratio of $a\% : b\% : c\%$, where $a\%$ of the data set is used to train the classifier, $b\%$ of the data set is used to train the attacker and the remaining $c\%$ of the data set is used to test the performance of the attacker. For the REDDIT-MULTI-12K and REDDIT-MULTI-5K data sets, we set $a = 90$, $b = 8$ and $c = 2$. As the size of the IMDB-MULTI data set is quite small, to have enough data for testing, we set $a = 50$, $b = 30$ and $c = 20$.

We compare the attacking performance of the proposed framework with the RL-S2V proposed in (Dai et al., 2018), random selection method and some variants of our proposed framework. We briefly describe these baselines: 1) **RL-S2V** is a reinforcement learning based attack framework (Dai et al., 2018), which allows adding and deleting edges to the graph with a fixed budget for all the graphs; 2) **Random** denotes an attacker that performs the proposed rewiring operations randomly; 3) **Random-s** is also based on random rewiring. Note that ReWatt can terminate before using all the budget. We record the actual number of rewiring actions made in our method and only allow the **Random-s** to take exactly the same number of rewiring actions as ReWatt; 4) **ReWatt-n** denotes a variant of the ReWatt, where the negative reward is fixed to $-0.5$ for all the graphs in the testing set; and 5) **ReWatt-a** is a variant of ReWatt, where we allow any nodes in the graph to be the third node $v_{thi_t}$ instead of only 2-hop neighbors.

As RL-S2V only allows a fixed budget for the all the graphs, when comparing to it, for ReWatt, we also fix the number of proposed rewiring operations to a fixed number $K$ for all the graphs. Note that a single proposed rewiring operation involves two edges, thus, for a fair comparison, we allow the RL-S2V to take $2K$ actions (adding/deleting edges). We set $K = 1, 2, 3$ in the experiments. To compare with the random selection method and the variants of ReWatt, we use flexible budget, more specially, we allow at most $p \cdot |\mathcal{E}_i|$ proposed rewiring operations for graph $G_i$. Here, $p$ is a fixed percentage and we set it to $p = 1\%, 2\%, 3\%$ in our experiments. We use the success rate as measure to evaluate the performance of the attacker. A graph is said to be successfully attacked if its label is changed when it is modified within the given budget.

The results are shown in Table 1. From the table, we can make the following observations: 1) Compared to RL-S2V, ReWatt can perform more effective attacks. Especially, in the IMDB-MULTI

| | REDDIT-MULTI-12K | | | REDDIT-MULTI-5K | | | IMDB-MULTI | | |
|---|---|---|---|---|---|---|---|---|---|
| K | 1 | 2 | 3 | 1 | 2 | 3 | 1 | 2 | 3 |
| ReWatt | 14.4% | 21.6% | 23.4% | 8.99% | 16.9% | 18.0% | 23.0% | 23.3% | 23.3% |
| RL-S2V | 9.46% | 18.5% | 21.1% | 4.49% | 16.9% | 18.0% | 2.00% | 6.00% | 3.33% |
| p | 1% | 2% | 3% | 1% | 2% | 3% | 1% | 2% | 3% |
| ReWatt | 25.2% | 32.9% | 38.7% | 11.2% | 20.2% | 27.0% | 23.0% | 23.0% | 23.3% |
| ReWatt-a | 26.1% | 35.1% | 42.8% | 5.60% | 21.3% | 30.3% | 24.3% | 25.0% | 25.6% |
| ReWatt-n | 17.6% | 25.7% | 31.1% | 5.60% | 14.6% | 19.1% | 21.3% | 21.3% | 21.6% |
| random | 10.3% | 15.7% | 21.6% | 3.30% | 12.4% | 16.9% | 1.33% | 1.33% | 1.66% |
| random-s | 6.30% | 6.70% | 9.45% | 5.60% | 6.74% | 11.0% | 1.00% | 1.33% | 1.66% |

Table 1: Performance comparison in terms of success rate

data set, where ReWatt outperforms the RL-S2V with a large margin; 2) ReWatt outperforms the Random method as expected. Especially, ReWatt is much more effective than Random-s which performs exactly the same number of proposed rewiring operations ReWatt. This also indicates that the Random method uses more rewiring operations for successful attacking than ReWatt; 3) The variant ReWatt-a outperforms ReWatt, which means if we do not constraint the rewiring operation to 2-hop neighbors, the performance of ReWatt can be further improved. However, as we discussed in earlier sections, this may lead to more "noticeable" changes of the graph structure; and 4) ReWatt-n performs worse than our ReWatt, which shows the advancement of using a flexible reward design.

## 5.2 ATTACKER ANALYSIS

In this subsection, we carry out experiments to analyze how ReWatt's change in graph structure affects the graph representation $\mathbf{u}$ calculated by eq. equation 2 and the logits $\mathbf{P}$ (the output immediately after the softmax layer of the classifier). For convenience, we denote the original graph as $G^o$ and the attacked graph as $G^a$ in this section. Correspondingly, the graph representation and logits for the original (attacked) graph are denoted as $\mathbf{u}^o$ ($\mathbf{u}^a$) and $\mathbf{P}^o$ ($\mathbf{P}^a$), respectively. To measure the difference in graph representation, we used the relative difference in terms of 2-norm defined as $RC(\mathbf{u}^o, \mathbf{u}^a) = \frac{\|\mathbf{u}^a - \mathbf{u}^o\|_2}{\|\mathbf{u}^o\|_2}$. The logits denote the probability distribution that the given graph belongs to each of the classes. Thus, we use the KL-divergence Kullback (1997) to measure the difference between the logits of the original and attacked graphs $KL(\mathbf{P}^o, \mathbf{P}^a) = \sum_{i=1}^{C} \mathbf{P}^o[i] \log \left( \frac{\mathbf{P}^o[i]}{\mathbf{P}^a[i]} \right)$, where $C$ is the number of classes in the data set and $\mathbf{P}[i]$ denotes the logit for the $i$-th class.

We perform the experiments on the REDDIT-MULTI-12K data set under the setting of allowing at most $3\% \cdot |\mathcal{E}|$ proposed rewiring operations. The results for the graph representation and logits are shown in Figure 1 and Figure 2, respectively. The graphs in the testing set are separated in two groups, one group contains all the graphs successfully attacked by ReWatt (shown in Figure 1a and Figure 2a), and the other one contains those survived from ReWatt's attack (shown in Figure 1b and Figure 2b). Note that, for comparison, we also include the results of Random-s on these two groups of graphs. In these figures, a single point represents a testing graph, the x-axis is the ratio $\frac{M}{|\mathcal{E}|}$, where $M$ is the number of rewiring operations ReWatt used before the attacking process terminating. Note that $M$ can be smaller than the budget as the process terminates once the attack successes.

As we can observe from the figures, compared with the Random-s, ReWatt can make more changes to both the graph representation and logits, using exactly the same number of proposed rewiring operations. Comparing Figure 1a with Figure 1b, we find that the perturbation generated by ReWatt affects the graph representation a lot even when it fails to attack the graph. This means our attack is perturbing the graph structure in a right way to fool the classifier, although it fails potentially due to the limited budget. Similar observation can be made when we compare Figure 2a with Figure 2b.

## 6 RELATED WORK

In recent years, adversarial attacks on deep learning models have attracted increasing attention in the area of computer vision. Many deep models are found to be easily fooled by adversarial samples, which are generated by adding deliberately designed unnoticeable perturbation to normal im-

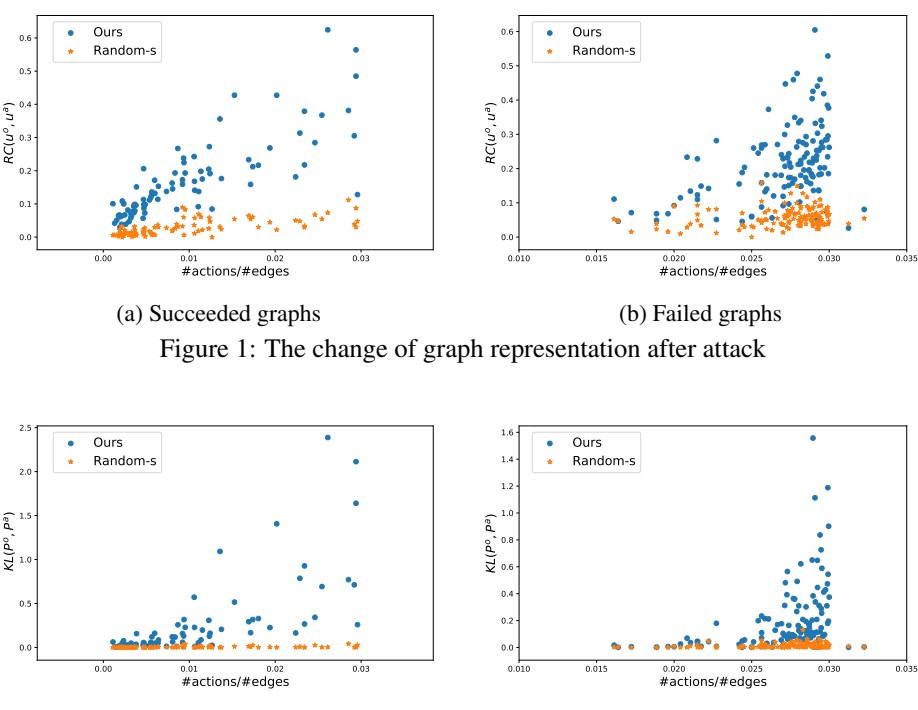

(a) Succeeded graphs

(b) Failed graphs

Figure 1: The change of graph representation after attack

(a) Succeeded graphs

(b) Failed graphs

Figure 2: The change of logits after attack

ages (Szegedy et al., 2013; Goodfellow et al., 2014). More algorithms with different level access to the target classifier have been proposed, including white-box attack models, which have access to the gradients (Moosavi-Dezfooli et al., 2016; Kurakin et al., 2016; Carlini & Wagner, 2017) and black-box attack model, which have limited access to the target classifier (Chen et al., 2017; Cheng et al., 2018; Ilyas et al., 2018).

Most of the aforementioned works are focusing in the computer vision domain, where the data sample can be represented in the continues space. Few attention has been payed into the discrete data structure such as graphs. Graph Neural Networks have been shown to bring impressive advancements to many different graph related tasks such as node classification and graph classification. Recent researches show that the graph neural networks are also venerable to adversarial attacks. (Zügner et al., 2018) proposed a greedy algorithm to perform adversarial attack to node classification task. Their algorithm tries to change the label of a target node by modifying both the graph structure and node features. (Dai et al., 2018) proposed a deep reinforcement learning based attacker to attack both the node classification and the graph classification task. (Zügner & Günnemann, 2019) designed an algorithm to impair the overall performance of node classification based on meta learning. All the three mentioned methods modify the graph structure by adding or deleting edges. A more recent work Wang et al. (2018) on attacking node classifications proposed to modify the graph structure by adding fake nodes. In this work, we propose to modify the graph structure using rewiring, which is shown to make less noticeable changes to the graph structure.

## 7 CONCLUSION

In this paper, we proposed a graph rewiring operation, which affect the graph structure in a less noticeable way than adding/deleting edges. The rewiring operation preserves some basic graph properties such as number of nodes and number of edges. We then designed an attacker ReWatt based on the rewiring operations using reinforcement learning. Experiments in 3 real world data sets show the effectiveness of the proposed framework. Analysis on how the graph representation and logits change while the graph being attacked provide us with some insights of the attacker.

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

## A  GRAPH LAPLACIAN BASED MEASURES

Many important graph properties are based on the eigenvalues of the Laplacian matrix of a graph (Chan & Akoglu, 2016). Here we list few:

- **Algebraic Connectivity** The algebraic connectivity of a graph $G$ is the second-smallest eigenvalue of its Laplacian matrix (Fiedler, 1973). Note that we only consider connected graphs in this work, so it is always larger than $0$. The larger the algebraic connectivity is, the more difficult it is to separate the graph into components (i.e., more edges need to be removed). The algebraic connectivity has previously been applied to measure network robustness Sydney et al. (2013).

- **Effective Graph Resistance** The effective graph resistance is a graph measure derived from the field of electric circuit analysis, where it is defined as the summation of effective resistance over all node pairs (Ellens et al., 2011). The effective graph resistance can be represented using the eigenvalues of Laplacian matrix as follows (Ellens et al., 2011)

$$R_e = |\mathcal{V}| \cdot \sum_{i=2}^{|\mathcal{V}|} \lambda_i.  \tag{9}$$

By Corollary 2, we can represent the change of the algebraic connectivity $\lambda_2$ as:

$$\Delta\lambda_2 = (2\mathbf{x}_2[fir] - \mathbf{x}_2[thi] - \mathbf{x}_2[sec])(\mathbf{x}_2[sec] - \mathbf{x}_2[thi]) \tag{10}$$

According to the above discussion, $\Delta\lambda_2$ is expected to be smaller for the operation of rewiring to 2-hop neighbor. Thus, the rewiring to 2-hop neighbor operation is expected to perturb the algebraic connectivity less compared with adding an edge between two nodes that are far away from each other. A similar argument can be built for effective graph resistance.

## B    PROOF OF COLLARY 1

**Corollary 2.** *For a given graph $G$ with Laplacian matrix $\mathbf{L}$, one proposed rewiring operation* $(v_{fir}, v_{sec}, v_{thi})$ *affects the eigen-value $\lambda_i$ by $\Delta\lambda_i$, for $i = 1, \ldots, |\mathcal{V}|$, where*

$$\Delta\lambda_i = (2\mathbf{x}_i[fir] - \mathbf{x}_i[thi] - \mathbf{x}_i[sec])(\mathbf{x}_i[sec] - \mathbf{x}_i[thi]) \tag{11}$$

*where $\mathbf{x}_i[index]$ denotes the $index$-th value of the eigenvector $\mathbf{x}_i$.*

*Proof.* Let $\Delta\mathbf{L}$ denotes the change in the Laplacian matrix after applying the rewiring operation $(v_{fir}, v_{sec}, v_{thi})$ to graph $G$. Then we have $\Delta\mathbf{L}[fir, sec] = \Delta\mathbf{L}[sec, fir] = 1$, $\Delta\mathbf{L}[fir, thi] = \Delta\mathbf{L}[thi, fir] = -1$, $\Delta\mathbf{L}[sec, sec] = -1$, $\Delta\mathbf{L}[thi, thi] = 1$ and 0 elsewhere. Thus

$$\begin{aligned}
\Delta\lambda_i &= \mathbf{x}_i^T \Delta\mathbf{L} \mathbf{x}_i \\
&= 2\mathbf{x}_i[fir]\mathbf{x}_i[sec] - \mathbf{x}_i[sec]^2 + \mathbf{x}_i[thi]^2 - 2\mathbf{x}_i[fir]\mathbf{x}_i[thi] \\
&= \mathbf{x}_i[thi]^2 - \mathbf{x}_i[sec]^2 + 2\mathbf{x}_i[fir](\mathbf{x}_i[sec] - \mathbf{x}_i[thi]) \\
&= (2\mathbf{x}_i[fir] - \mathbf{x}_i[thi] - \mathbf{x}_i[sec])(\mathbf{x}_i[sec] - \mathbf{x}_i[thi])
\end{aligned}$$

which completes the proof. $\qquad\square$

## C    EMPIRICAL INVESTIGATION OF THE REWIRING OPERATION

In this section, we conduct experiments to empirically show the advancements of the proposed rewiring operator compared with the adding/deleting edge operator. We compare them from two perspectives: 1) connectivity after the attack and 2) change in eigenvalues after the attack. The experiments are carried out on the REDDIT-MULTI-12K dataset. On each of the graph successfully attacked by ReWatt, we perform exactly the same number of deleting/adding edge operator on it. For connectivity, the average number of components in the clean graphs is 2.6, this number becomes 3.02 after the rewiring attack while it becomes 5.2 after the deleting/adding edges attack. On the other hand, only 20% of the graphs get more disconnected (having more components) after ReWattattack than the original ones, while 87% of the graphs get more disconnected after the adding/deleting edges attack. Clearly, the rewiring operator is less likely to disconnect the graph. The comparison of the change in eigenvalues is shown in Figure 3, where we compare the change in different eigenvalues of the graphs after these two attacks. Specifically, we first compute the average relative change in the $i$-th eigenvalue after both attacks as follows:

$$r_{\lambda_i} = \frac{|\lambda_i^{ori} - \lambda_i^{attack}|}{\lambda_i^{ori}}, \tag{12}$$

where $\lambda_i^{ori}$ denotes the $i$-th eigenvalue of the clean graph while $\lambda_i^{attack}$ denotes the $i$-th eigenvalue of the attacked graph. We then take the average of the above value over all the succeeded graphs, which we denoted as $\bar{r}_{\lambda_i}$. Specially, we use $\bar{r}_{\lambda_i}^{re}$ to denote the average change ratio after $ReWatt$ while using $\bar{r}_{\lambda_i}^{d/a}$ to denote the average change ratio after deleting/adding edge attack. To compare the these two attacks, we calculate $\bar{r}_{\lambda_i}^{d/a}/\bar{r}_{\lambda_i}^{re}$ and the results are shown in Figure 3. The results show that in most of the cases, the deleting/adding edges attack makes much more changes to the eigenvalues as the value $\bar{r}_{\lambda_i}^{d/a}/\bar{r}_{\lambda_i}^{re}$ is way larger than 1.

By conducting these two experiments, we empirically conclude that the proposed re-wiring operator makes more subtle changes to graphs than existing methods.

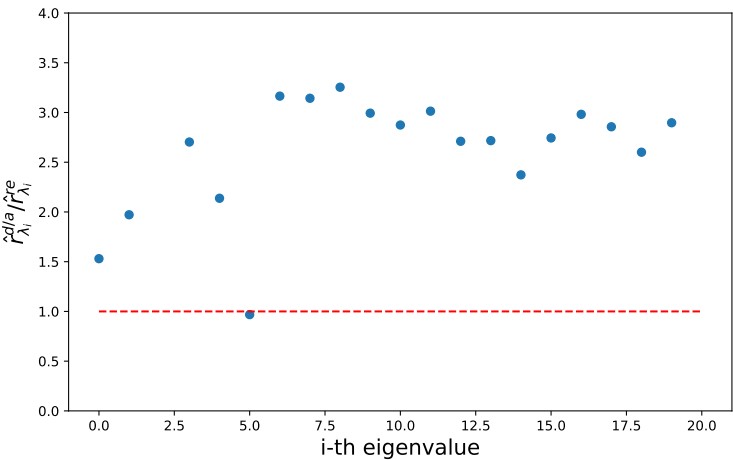

Figure 3: Comparison in the change of eigenvalues

## D STATISTICS OF THE DATASETS

The statistics of the datasets are given in Table 2. In this table, #nodes denotes the average number of nodes over all graphs and #edges denotes the average number of edges over all graphs. ACC denotes the mean of Average Clustering Coefficient (ACC) over all graphs. GCC denotes the mean of Global Clustering Coefficient (GCC) over all graphs.

|  | #graphs | #labels | #nodes | #edges | ACC | GCC |
|---|---|---|---|---|---|---|
| REDDIT-MULTI-12K | 11,929 | 12 | 391.41 | 456.89 | 0.0331 | 0.0087 |
| REDDIT-MULTI-5K | 4,999 | 5 | 508.52 | 594.87 | 0.0268 | 0.0038 |
| IMDB-MULTI | 1,500 | 3 | 13 | 65.94 | 0.968 | 0.8955 |

Table 2: Statistics of the data sets

