# OpenReview forum: "Attacking Graph Convolutional Networks via Rewiring"
_ICLR.cc/2020/Conference — Reject_

### Official Review · AnonReviewer2 · 2019-10-23
**Official Blind Review #2**

**Rating:** 6

**Review:**

In this paper, the authors studied the adversarial attack problem for graph classification problem with graph convolutional networks. After observing that traditional attack by adding or deleting edges can change graph eigenvalues, the author proposed to attack by adding rewiring operation which make less effects. Rewiring does not change the graph edge number and the average degree. Further, the authors propose an RL based learning method to learn the policy of doing rewiring operation. Experiments show that the proposed method can make more successful attack on social network data than baselines and previous methods.

The idea of using rewiring to make graph attack is interesting and sensible. The proposed RL-based method where the search space is constraint also can solve the problem. However, I have a few concerns on the experiments.

1. In figure 3, the authors also show that the proposed method can make less noticeable changes on eigenvalue. But are these changes still noticeable compared to original one? Please also show these information.
2. 2% data for testing is too few for me. The authors should increase these number. In addition, how many replication of experiments did the author do? The author should give the variance of the results and make significant test if needed.
3. What is the prediction accuracy of the target classifier? Did the attacker flip more correct predictions?


**Experience Assessment:**

I have published one or two papers in this area.

**Review Assessment: Checking Correctness Of Derivations And Theory:**

I assessed the sensibility of the derivations and theory.

**Review Assessment: Checking Correctness Of Experiments:**

I assessed the sensibility of the experiments.

**Review Assessment: Thoroughness In Paper Reading:**

I read the paper at least twice and used my best judgement in assessing the paper.

---

> ### Author Response · Authors · 2019-11-15
> **Response to Official Blind Review #2**
>
> Thank you for the valuable comments and suggestions.
>
> We address the concerns from the reviewer as follows:
>
> Q1: In figure 3, the authors also show that the proposed method can make less noticeable changes on eigenvalue. But are these changes still noticeable compared to original one? Please also show these information.
>
> A1: In Figure 3, we have compared the changes made by the rewiring attack and the random adding/deleting operation. Here, we directly provide the changes. After the rewiring attack performed by ReWatt, the eigenvalues of the Laplacian matrix change about 2.6% on average, while they change about 7.78% after random adding/deleting attack.
>
> Q2: 2% data for testing is too few for me. The authors should increase these number. In addition, how many replication of experiments did the author do? The author should give the variance of the results and make significant test if needed.
>
> A2: For the attack experiments,  we have to split each dataset into three non-overlapping parts: 1) a classifier-training set to train the classifier to be attacked; 2) an attacker-training set to train the attacker; and 3) an attacker-testing set to test the performance of the trained attacker. To test the performance of the attack performance, we need to obtain a well-trained GCN; as a result, we need a large portion of each dataset (or a classifier-training set) to train the GCN algorithm. In REDDIT-MULTI-12K and REDDIT-MULTI-5K, we use $90\%$ of the entire dataset to train the classifier. Furthermore, the remaining $10\%$ of the dataset is used to train and test the attacker, where $80\%$ of the remaining data is used as the attacker-training set, while $20\%$ of the remaining data is used as the attacker-testing set. Hence, the ratio between the attacker-training and attacker-testing sets is 4:1 which suggests the attacker-training set and attacker-testing set are well balanced. Although we could use a larger portion for attacker-testing set by reducing the classifier-training set, it affects the performance of the classifier to be attacked. In the IMDB-MULTI dataset, to have enough graphs to train and test the attacker, we compromise the performance of the classifier by using only $50\%$ of the entire dataset to train the classifier.
>
> To compare ReWatt with RL-S2V, we run these two methods on $5$ different data splits and report the average performance with variance. Specifically, for each split, we keep the attacker-training set fixed to make sure the being attacked classifier is the same over different runs. We then randomly shuffle the remaining dataset and split it into the attacker-training set and the attacker-testing set. The performance on REDDIT-MULTI-12K is as follows:
>                         1                        2                         3
> RL-S2V : 0.09115 (0.0262); 0.1677 (0.0225); 0.2048 (0.0106)
> ReWatt:  0.1271 (0.0251);  0.2115 (0.0194); 0.2503 (0.0148)
>
> where the standard deviation is shown in parenthesis.
> We also did the significance test, where the $p$-values for the three settings (1, 2, 3 rewiring operations) are $0.032$; $0.0066$; $0.00149$ respectively. So, ReWatt performs significantly better than RL-S2V.
>
> On the REDDIT-MULTI-5K dataset, the performance is as follows
>                         1                        2                         3
> RL-S2V : 0.0486 (0.0064); 0.1612 (0.0236); 0.1948 (0.0128)
> ReWatt:  0.0711 (0.0173);  0.1612 (0.0236); 0.1912 (0.0193)
> The $p$-values are $0.1015$, $1.0$, $0.797$ respectively. So, RL-S2V and ReWatt do not perform significantly different on REDDIT-MULTI-5K.
>
> On the IMDB-MULTI dataset, the performance is as follows
>                         1                        2                         3
> RL-S2V : 0.024 (0.0063); 0.059 (0.0068); 0.0624 (0.0284)
> ReWatt:  0.2306 (0.0149);  0.233 (0.0164); 0.2338 (0.0178)
> The  $p$-values are all smaller than $0.00001$. Hence, ReWatt significantly outperforms RL-S2V on IMDB-MULTI datset.
>
>
> Q3: What is the prediction accuracy of the target classifier? Did the attacker flip more correct predictions?
>
> A3: We take the REDDIT-MULTI-12K dataset as an example to answer this question. The prediction accuracy of the target classifier on the original (unattacked) testing set is $43.24\%$, after the attack, the accuracy is reduced to $32.88\%$. According to this observation, the attacker flips more correct predictions than incorrect predictions.

---

### Official Review · AnonReviewer1 · 2019-10-23
**Official Blind Review #1**

**Rating:** 3

**Review:**

This paper proposes the ReWatt method to attack graph classification models by making unnoticeable perturbations on graph. Reinforcement learning was leveraged to find a rewiring operation a = (v1; v2; v3) at each step, which is a set of 3 nodes. In the first step, an existing edge (v1, v2) in the original graph is selected and removed. Then another node v3 that is 2-hop away from v1 and not 1-hop away is selected.  Finally (v3, v1) is connected as a new edge. Some analysis shows that the rewiring operation tends to make smaller changes to the eigenvalues of the graph's Laplacian matrix compared with simply adding and deleting edges, making it difficult to detect the attacks.

Pros

1. The rewiring operation is more unnoticeable. Small change is shown on the eigenvalues with one rewiring operation.

2. The proposed ReWatt method is effective in attacking the graph classification algorithm, facilitated by the policy network to pick the edges.

3. ReWatt outperforms the RL-S2V in terms of success rate, especially when the second step in the rewiring process is not limited by 2-hops away from v1.

4. The paper measured the relative difference between the graph embeddings in terms of L2 norm and measured the KL-divergence in probabilities.

Cons

1. It's quite surprising that ReWatt achieves higher success rate than RL-S2V (first two rows of Table 1).  RL-S2V considers a properly larger set of attacks and uses Q-learning (in contrast to actor critic in ReWatt).  So is it the conclusion that actor critic is better than Q-learning?  Perhaps it will be illustrative to experiment with replacing Q-learning in RL-S2V by actor critic.  This can be implemented in the framework of ReWatt: in Eq 5, replace $p_{fir} * p_{thi}$ by $p(add/remove | e_t)$.

2. The attack is specifically designed for graph classification, while the graph convolutional filter is widely used in other problems like node classification and link prediction. Can it be applied to such problems as well?

3. In addition to RL-S2V, it will be helpful to compare with Nettack (Z¨ugner et. al, 2018).  It employs an admissible set of perturbations, which can be adapted for the rewiring attack.

4. The paper shows the change of eigenvalues under one rewiring operation. How does it change after multiple operations?  In addition, the smaller change to the eigenvalues is compared with rewiring to more distant nodes or adding an edge between two distant nodes.  That is, it is under a *given* $v_{fir}$ and $v_{sec}$.  A different attack may select a different $v_{fir}$ and $v_{sec}$ in the first place.  So it is still not clear whether rewiring leads to less noticeable changes.

5. The experiment splits the dataset into three parts, training set, rewiring operation set, and test set. However, for those predicted incorrectly on the rewiring operation set, the success rate should not be counted.  Perhaps this is already done?

**Experience Assessment:**

I have published one or two papers in this area.

**Review Assessment: Checking Correctness Of Derivations And Theory:**

I assessed the sensibility of the derivations and theory.

**Review Assessment: Checking Correctness Of Experiments:**

I assessed the sensibility of the experiments.

**Review Assessment: Thoroughness In Paper Reading:**

I read the paper at least twice and used my best judgement in assessing the paper.

---

> ### Author Response · Authors · 2019-11-15
> **Response to Official Blind Review #1--Part 1**
>
> Thank you for the valuable comments and suggestions.
>
> We address the concerns from the reviewer as follows:
> Q1: It's quite surprising that ReWatt achieves higher success rate than RL-S2V (first two rows of Table 1).  RL-S2V considers a properly larger set of attacks and uses Q-learning (in contrast to actor-critic in ReWatt).  So is it the conclusion that actor-critic is better than Q-learning?  Perhaps it will be illustrative to experiment with replacing Q-learning in RL-S2V by actor-critic.  This can be implemented in the framework of ReWatt: in Eq 5, replace $p_{fir}*p_{thi}$ by $p(add/remove|e_t)$.
>
> A1: We agree that RL-S2V has a larger attack space, which means the optimal solution it can achieve is as good or better than the one our method can find. However, both methods are not guaranteed to always find the optimal solution in the given attack space. We list some potential reasons to explain why ReWatt can outperform RL-S2V as follows:
> 1) When performing an adding/deleting edge action in RL-S2V, it chooses two nodes sequentially. Then it decides to add an edge between two nodes if they are not connected, otherwise, the edge between them is removed. Since most graphs are very sparse, the RL-S2V algorithm is, by design, biased to adding an edge. On the other hand, ReWatt removes an edge and then add another edge. The adding/deleting edge operations are more balanced.
> 2) The reward design in ReWatt is different from RL-S2V. In RL-S2V, a non-zero reward is only given at the end of an attacking session. Specifically, at the end of an attacking session, a positive reward of $1$ is given if the attack succeeded, otherwise a negative reward $-1$ is given. All the intermediate steps get $0$ reward. In ReWatt, the reward is given after each action. A positive reward is given once an action leads to a successful attack. A negative reward is penalized to take each action if it does not directly lead to a successful attack, which encourages the attacker to make as few actions as possible. Furthermore, we also proposed an adaptive negative reward design, which determines the value of the negative reward according to the size of each graph. In fact, the design of this adaptive negative reward has shown to be very effective and important to the ReWatt framework. As shown in Table 1, ReWatt-n (which is a variant of ReWatt without adaptive negative reward design) performs much worse than ReWatt. Specifically, if we apply ReWatt-n in the same setting of RL-S2V (with fixed actions), its performance is not as good as RL-S2V in REDDIT-MULTI-12K and REDDIT-MULTI-5K datasets. The performance of ReWatt-n on REDDIT-MULTI-12K is [11.26%; 14.7%; 18.02] while RL-S2V achieves [9.46; 18.5% 21.1%]. On the REDDIT-MULTI-5K, the performance of ReWatt-n is [4.49%; 5.62%; 6.74%] while RL-S2V archives [4.49%; 16.9%; 18.0%]. Hence, the design of adaptive negative reward could be an important reason why ReWatt can perform better than RL-S2V.
>
> Also, please note that RL-S2V cannot be implemented with actor-critic by simply replacing $p_{fir}*p_{thi}$ with $p(add/remove|e_t)$ in the framework of ReWatt. This is because the action of ReWatt is different from RL-S2V as described in 1). The edge $e_t$ chosen by ReWatt is an existing edge in the graph, therefore we can only delete it from the graph and can not add it to the graph. Hence, $p(add/remove|e_t)$ cannot be performed in practice.
>
> Q2: The attack is specifically designed for graph classification, while the graph convolutional filter is widely used in other problems like node classification and link prediction. Can it be applied to such problems as well?
>
> A2: The ReWatt framework can be applied to attack node level tasks such as node classification and link prediction by adjusting the design of the rewards. For example, for node classification, we can design the reward based on the overall performance of the targeted classifier. Specifically, if the goal is to decrease the overall performance of a node classification classifier, a positive reward can be given when an action reduces the overall performance (evaluated on a validation set) and a negative reward can be given if an action increases the accuracy.
>
> Q3: In addition to RL-S2V, it will be helpful to compare with Nettack (Z¨ugner et. al, 2018). It employs an admissible set of perturbations, which can be adapted for the rewiring attack.
>
> A3: Our work focuses on the graph-level attack, while Nettack is designed for targeted node-level attack. It is not straightforward to adapt Nettack for graph-level tasks. Hence, we didn’t compare our method with Nettack. However, we do agree that some of the constraints used in Nettack can be incorporated into our framework, which can be a promising future step to make the attack even more unnoticeable.

---

> > ### Author Response · Authors · 2019-11-15
> > **Response to Official Blind Review #1--Part 2**
> >
> > Q4: The paper shows the change of eigenvalues under one rewiring operation. How does it change after multiple operations?  In addition, the smaller change to the eigenvalues is compared with rewiring to more distant nodes or adding an edge between two distant nodes.  That is, it is under a *given* $v_{fir}$  and $v_{sec}$.  A different attack may select a different  $v_{fir}$  and $v_{sec}$ in the first place.  So it is still not clear whether rewiring leads to less noticeable changes.
> >
> > A4: Applying multiple rewiring operations to a graph can be viewed as applying these operations one by one. So, in the worst case, the changes can be accumulated. In some specific cases, the changes made by multiple rewiring operations can be smaller than direct accumulation. For example, the two rewiring operations $(v_1,v_2,v_3)$ and $(v_1, v_3, v_4)$ can be merged to one single rewiring operation $(v_1,v_2,v_4)$. Note that the experiments in Appendix C are not based on a single rewiring operation but potentially multiple rewiring operations. So, we have empirically shown that even with multiple rewiring operations, the change to the eigenvalues is still small. We have empirically shown in Appendix C that, with the same number of operations, ReWatt made smaller changes to the eigenvalues of the Laplacian matrix than random adding/deleting operation.
> >
> > Q5: The experiment splits the dataset into three parts, training set, rewiring operation set, and test set. However, for those predicted incorrectly on the rewiring operation set, the success rate should not be counted.  Perhaps this is already done?
> >
> > A5:  Each dataset is split into three non-overlapping parts: 1) a classifier-training set to train the classifier to be attacked; 2) the attacker-training set to train the attacker; and 3) the attacker-testing set to test the performance of the trained attacker. So, the attacker learns to perform the rewiring operation properly on the attacker-training set and then attacks the attacker-testing set by performing rewiring operations. The success rate reported in the paper is only based on the attacker-testing set.

---

### Official Review · AnonReviewer4 · 2019-10-28
**Official Blind Review #4**

**Rating:** 3

**Review:**

This paper proposes a new type of adversarial attack setting for graphs, namely graph rewiring operation, which deletes an edge in the graph and adds a new edge between one node of the first edge and one of its 2-hop neighbors. This new attack is proposed to make the perturbations unnoticeable compared with adding or deleting arbitrary edges. To solve this problem, a reinforcement learning based approach is proposed to learn the attack strategy in the black-box manner. Experiments conducted on several datasets prove the effectiveness of the proposed with over an existing method and baseline methods.

Overall, this paper proposes a new adversarial setting for graphs to make the modifications unnoticeable. A reinforcement learning method is proposed to generate adversarial examples under the proposed setting. The writing is clear. However, I have several concerns about this paper as follows.

1. The proposed graph rewiring operation is a special operation of the general adding and deleting operations (i.e., rewiring is operated as deleting an edge and adding a new edge with some constrains). The motivation of using rewiring is to make the perturbations unnoticeable. Besides presenting the theoretical results on this property of the rewiring operation, it's better to provide some empirical results (e.g., generated adversarial graphs) to prove that the rewiring operation can make the adversarial graphs unnoticeable in practice.

2. In Table 1, why are the results of ReWatt better than RL-S2V? Since there are more constrains (i.e., smaller action space) in ReWatt than RL-S2V, RL-S2V could be easier to fool GCNs. The authors could explain more on the results.

3. What are the differences between the proposed attack method based on reinforcement learning and the method in RL-S2V? RL-S2V is also based on reinforcement learning. The authors should clearly introduce the novelty of the proposed method as well as the contributions.

**Experience Assessment:**

I have published one or two papers in this area.

**Review Assessment: Checking Correctness Of Derivations And Theory:**

I carefully checked the derivations and theory.

**Review Assessment: Checking Correctness Of Experiments:**

I carefully checked the experiments.

**Review Assessment: Thoroughness In Paper Reading:**

I read the paper thoroughly.

---

> ### Author Response · Authors · 2019-11-15
> **Response to Official Blind Review #4**
>
> Thank you for the valuable comments and suggestions.
>
> We address the concerns from the reviewer as follows:
>
> Q1: The motivation for using rewiring is to make the perturbations unnoticeable. Besides presenting the theoretical results on this property of the rewiring operation, it's better to provide some empirical results (e.g., generated adversarial graphs) to prove that the rewiring operation can make the adversarial graphs unnoticeable in practice.
>
> A1: We have performed empirical investigations of the rewiring operation, which can be found in Appendix C. In summary, the rewiring attack performed by ReWatt does smaller changes to the attacked graph in terms of connectivity and the Laplacian spectrum. Furthermore, as requested by Blind Reviewer #3, we have done some experiments to show the rewiring attack performed by ReWatt also does small changes to the spectrum of the adjacency matrix and distribution of the edge_centrality (please see the responses to Q1 and Q2 of Blind Review #3 ).
>
> Q2: In Table 1, why are the results of ReWatt better than RL-S2V? Since there are more constraints (i.e., smaller action space) in ReWatt than RL-S2V, RL-S2V could be easier to fool GCNs. The authors could explain more about the results.
>
> A2: We agree that RL-S2V has a larger action space, which means the optimal solution it can achieve is as good or better than the one our method can find. However, both methods are not guaranteed to always find the optimal solution in the given action space. We list some potential reasons to explain why ReWatt can outperform RL-S2V as follows:
> 1) When performing an adding/deleting edge action in RL-S2V, it chooses two nodes sequentially. Then it decides to add an edge between two nodes if they are not connected, otherwise, the edge between them is removed. Since most graphs are very sparse, the RL-S2V algorithm is, by design, biased to adding an edge. On the other hand, ReWatt removes an edge and then add another edge. The adding/deleting edge operations are more balanced.
> 2) The reward design in ReWatt is different from RL-S2V. In RL-S2V, a non-zero reward is only given at the end of an attacking session. Specifically, at the end of an attacking session, a positive reward of $1$ is given if the attack succeeded, otherwise a negative reward $-1$ is given. All the intermediate steps get $0$ reward. In ReWatt, the reward is given after each action. A positive reward is given once an action leads to a successful attack. A negative reward is penalized to take each action if it does not directly lead to a successful attack, which encourages the attacker to make as few actions as possible. Furthermore, we also proposed an adaptive negative reward design, which determines the value of the negative reward according to the size of each graph. In fact, the design of this adaptive negative reward has shown to be very effective and important to the ReWatt framework. As shown in Table 1, ReWatt-n (which is a variant of ReWatt without the adaptive negative reward design) performs much worse than ReWatt. Specifically, if we apply ReWatt-n in the same setting of RL-S2V (with fixed actions), its performance is not as good as RL-S2V in REDDIT-MULTI-12K and REDDIT-MULTI-5K datasets. The performance of ReWatt-n on REDDIT-MULTI-12K is [11.26%; 14.7%; 18.02] while RL-S2V achieves [9.46; 18.5% 21.1%]. On the REDDIT-MULTI-5K, the performance of ReWatt-n is [4.49%; 5.62%; 6.74%] while RL-S2V archives [4.49%; 16.9%; 18.0%]. Hence, the design of our adaptive negative reward could be an important reason why ReWatt can perform better than RL-S2V.
>
> Q3: What are the differences between the proposed attack method based on reinforcement learning and the method in RL-S2V? RL-S2V is also based on reinforcement learning. The authors should clearly introduce the novelty of the proposed method as well as the contributions.
>
> A3: A major contribution is that we propose to use rewiring to perform the attack. We also show that the rewiring operation is less noticeable both theoretically and empirically. On the other hand, the architecture of the reinforcement framework of ReWatt is also different from RL-S2V. We have stated the differences in the response to Q2

---

### Official Review · AnonReviewer3 · 2019-10-28
**Official Blind Review #3**

**Rating:** 6

**Review:**

The paper addresses a real problem.  Most attacks on graphs can be easily identified [1].  This paper argues that if one rewires the graph (instead of adding/deleting nodes/edges) such that the top eigenvalues of the Laplacian matrix are only slightly perturbed then the attacker can go undetected.

The paper should address the following issues:

1. There is no discussion on tracking the path capacity of the graph as measured by the largest eigenvalue of the adjacency matrix and the eigengaps between the largest in module eigenvalues of the adjacency matrix .  Rewiring often affects the path capacity even if one makes sure the degree distribution is the same and restricts the rewiring to 2-hop neighbors.

2. Rewiring affects edge centrality and so one needs to show that the proposed algorithm doesn't change the distribution over edge centrality.

3. In social networks, the highest eigenvalues of the adjacency matrix are very close to each other because of all the triangles.  The paper will be stronger if it included how the proposed method performs under various random graph models -- e.g., Gnp random graph, preferential attachment, and small-world.

Miscellaneous notes:

- The captions for the figures should be more informative.

- Table 2 should list more characteristics of the graphs such as number of nodes, number of edges, exponent of the degree distribution, global clustering coefficient, average clustering coefficient, diameter, average path length.

- "Zgner &Gnnemann" is misspelled.

- "As we can observed from the figures, ..." has a typo in it.

__________________________________________________
[1]  B. Miller, M. Çamurcu, A. Gomez, K. Chan, T. Eliassi-Rad. Improving Robustness to Attacks Against Vertex Classification. In The 15th International Workshop on Mining and Learning with Graphs (held in conjunction with ACM SIGKDD’19), Anchorage, AK, August 2019.


**Experience Assessment:**

I have published one or two papers in this area.

**Review Assessment: Checking Correctness Of Derivations And Theory:**

I carefully checked the derivations and theory.

**Review Assessment: Checking Correctness Of Experiments:**

I carefully checked the experiments.

**Review Assessment: Thoroughness In Paper Reading:**

I read the paper thoroughly.

---

> ### Author Response · Authors · 2019-11-15
> **Response to Official Blind Review #3**
>
> Thank you for the valuable comments and suggestions.
> Thanks for letting us know about the existence of another interesting paper in the field of adversarial attacks in the graph domain. We have cited it accordingly in the revision.
> We address the key concerns mentioned by the reviewer as follows:
>
> Q1: There is no discussion on tracking the path capacity of the graph as measured by the largest eigenvalue of the adjacency matrix and the eigengaps between the largest in module eigenvalues of the adjacency matrix.  Rewiring often affects the path capacity even if one makes sure the degree distribution is the same and restricts the rewiring to 2-hop neighbors.
> A1: We empirically verify that both the largest eigenvalue and the spectral gap of the adjacency matrix will not change too much after the rewiring attack performed by the ReWatt framework. We take the REDDIT-MULTI-12K as a representative dataset to conduct the verification experiments. Specifically, the experiments are conducted on the graphs (from the testing set) that are successfully attacked by ReWatt. Over this set of graphs, the mean of the largest eigenvalue of each initial graph (i.e., before the attack) is around 11.95. We calculate the change of the eigenvalue after rewiring attack by comparing with the original largest eigenvalue as follows:
>                                            $ |\lambda_{ori} - \lambda_{att}|$
> where $\lambda_{ori}$ denotes the original largest eigenvalue and $\lambda_{att}$ denotes the largest eigenvalue after rewiring attack. We then average this change over all the graphs in the set. On average, after the rewiring attack, the largest eigenvalue of each graph changes 0.042, which is quite small given the magnitude of the largest eigenvalue is around 11.95.
> On the other hand, the average spectral gap over the set of graphs is 0.1449. After the rewiring attack, the average spectral gap becomes 0.1204. The average change over the set of graphs is 0.049. Hence, the change in the spectral gap is also small.
>
> Q2: Rewiring affects edge centrality and so one needs to show that the proposed algorithm doesn't change the distribution over edge centrality.
> A2:  We conduct verification experiments in the same set of graphs as in response to Q1. For each graph in this set, we use the two-sample Kolmogorov-Smirnov Test to test whether the edge centrality values before and after attacking are from the same distribution. The null hypothesis of this test is that the two samples are from the same distribution. We are supposed to reject the null hypothesis when the p-value is small. When the p-value is large, we cannot reject the null hypothesis. The average p-value over all the graphs in the set is 0.568. 58% of the graphs are with $p$-value larger than 0.5. 31% of graphs are with $p$-value smaller than 0.05, which indicates the rejection of the null hypothesis. The remaining 11% of graphs have $p$-value between 0.05 and 0.5. So, the rewiring attack may affect the edge centrality distribution. However, empirically, for most of the graphs,  the edge centrality distribution of the attacked graph is not significantly different from the original one.
>
> Q3: In social networks, the highest eigenvalues of the adjacency matrix are very close to each other because of all the triangles.  The paper will be stronger if it included how the proposed method performs under various random graph models -- e.g., Gnp random graph, preferential attachment, and small-world.
> A3: The analysis (on the Laplacian spectrum) in the paper is for general graphs but not limited to social networks. The proposed framework is designed to attack the graph classification task. However,  there are no natural and meaningful labels associated with the random graphs and we cannot perform graph classification on random graphs.  A possible way is to construct labels while generating these graphs. However,  such synthetic labeling could have a great bias in the results depending on how the labels are selected. So, we do not apply the proposed framework to random graphs.
>
> Finally, thanks for providing the miscellaneous notes, we have updated most of them accordingly in the updated version of the paper. Due to the space limit, we do not include more information in the caption of the figures. They can be found in the text of the paper.

---

### Decision · Program_Chairs · 2019-12-19

**Decision:**

Reject

**Comment:**

This paper proposes a method for attacking graph convolutional networks, where a graph rewiring operation was introduced that affects the graph in a less noticeable way compared to adding/deleting edges. Reinforcement learning is applied to learn the attack strategy based on the proposed rewiring operation. The paper should be improved by acknowledging/comparing with previous work in a more proper way. In particular, I view the major innovation is on the rewiring operation and its analysis. The reinforcement learning formulation is similar to Dai et al (2018). This connection should be made more clear in the technical part. One issue that needs to be discussed on is that if you directly consider the triples as actions, the space will be huge. Do you apply some hierarchical treatment as suggested by Dai et al. (2018)?  The review comments should be considered to further improve too.